# Single and Joint Associations of Polycyclic Aromatic Hydrocarbon Exposure with Liver Function during Early Pregnancy

**DOI:** 10.3390/toxics11100863

**Published:** 2023-10-16

**Authors:** Mi Dai, Lei Luo, Caiyan Xie, Zhongbao Chen, Mingzhe Zhang, Yan Xie, Xuejun Shang, Xubo Shen, Kunming Tian, Yuanzhong Zhou

**Affiliations:** 1The Third Affiliated Hospital, Zunyi Medical University, Zunyi 563000, China; dm642350@163.com; 2School of Public Health, Zunyi Medical University, Zunyi 563000, China; dr_luolei7007@163.com (L.L.); ycx5224015525@163.com (C.X.); xie814yan@163.com (Y.X.); zy96_shenxubo@163.com (X.S.); 3Key Laboratory of Maternal & Child Health and Exposure Science, Guizhou Higher Education Institutes, School of Public Health, Zunyi Medical University, Zunyi 563000, China; 4Renhuai Center for Disease Control and Prevention, Zunyi 563000, China; chenzhongbao0128@163.com; 5Reproductive Center, Affiliated Hospital of Zunyi Medical University, Zunyi 563000, China; mingzhezhang825@163.com; 6Department of Andrology, School of Medicine, Jinling Hospital, Nanjing University, Nanjing 210002, China; shangxj98@sina.com

**Keywords:** polycyclic aromatic hydrocarbons, liver function, BKMR, pregnant women, Zunyi birth cohort

## Abstract

The individual and combined associations of polycyclic aromatic hydrocarbons (PAHs) metabolites on liver function during pregnancy are still lacking. We aimed to explore the connection between urinary PAH metabolites and liver function in early pregnant women in southwest China based on the Zunyi birth cohort. Ten urinary PAH metabolites and five liver function parameters during early pregnancy were measured. The associations of single PAHs with parameters of liver function were assessed using multiple linear regression. A Bayesian kernel machine regression (BKMR) model was used to evaluate the joint associations of the PAH mixture with outcomes. We found that each 1% increment of urinary 2-hydroxyphenanthrene (2-OH-PHE) was associated with 3.36% (95% CI: 0.40%, 6.40%) higher alanine aminotransferase (ALT) and 2.22% (95% CI: 0.80%, 3.67%) higher aspartate aminotransferase (AST). Each 1% increment in 1-hydroxy-phenanthrene (1-OH-PHE) was significantly associated with 7.04% (95% CI: 1.61%, 12.75%) increased total bile acid (TBA). Additionally, there was a significant positive linear trend between 2-OH-PHE and AST and 1-OH-PHE and TBA. BKMR also showed a significant positive association of PAH mixture with AST. Our results indicate that PAH metabolites were associated with increased parameters of liver function among early pregnant women. Early pregnant women should pay more attention to the adverse relationships between PAHs and liver function parameters to prevent environment-related adverse perinatal outcomes.

## 1. Introduction

During pregnancy, the hemodynamics of pregnant women profoundly changes. In early and late pregnancy, cardiac output increases by 30–40%, with a corresponding increase in total liver blood flow [1,2]. Meanwhile, during normal pregnancy, the levels of serum parameters of liver function are usually below the non-pregnant state due to the expansion of the extracellular fluid compartment [3]. Ample evidence indicates that increased serum liver transaminase in early pregnancy is closely associated with pregnancy complications and adverse embryo developmental outcomes [4]. Liver dysfunction often leads to pre-eclampsia, gestational diabetes, and an increased risk of ruptured variceal bleeding in pregnant women [5,6,7]. Moreover, the fetus is more likely to be premature or have a low birth weight [8]. Therefore, it is important to maintain normal liver function to defend the health of perinatal women and fetuses. In addition, a growing amount of evidence has pointed to various environmental contaminants as important contributors to aberrant liver function [9,10].

Polycyclic aromatic hydrocarbons (PAHs) are known as a class of widely used environmental compounds. Humans can be exposed to PAHs through skin contact, contaminated air inhalation, and consumption of contaminated food and beverages [11]. PAHs are carcinogenic and can cause damage to the endocrine system, reproductive system, liver, and kidney system [9,12,13,14]. Moreover, multiple studies have demonstrated the hepatoxicity of PAHs. For instance, high PAH levels increased the risk of non-alcoholic fatty liver disease in a US National Health and Nutrition Examination Survey study [15]. PAH exposure was also found to be associated with impaired liver function among garbage collectors in rural areas of Henan, China [16]. Xu et al. also found a positive association between 2-fluorene exposure and ALT levels in female adolescents [17]. However, the relationship between PAH exposure and liver function in pregnant women, especially during the first-trimester period which is the important window, is yet to be explored. 

Considering the critical role of liver function in the maintenance of perinatal health and embryonic development in early pregnancy, relevant studies are warranted to recognize the potential risk of PAHs for abnormal liver function. Therefore, we explored the individual and mixed relationships between PAH metabolites and parameters of liver function based on the Zunyi birth cohort. Multiple linear regressions were used to investigate the individual relationships between urinary PAH metabolites and parameters of liver function, including alanine aminotransferase (ALT), aspartate aminotransferase (AST), alkaline phosphatase (ALP), total bile acid (TBA), and AST/ALT. Furthermore, given the joint exposure of PAHs in the real world, we also conducted Bayesian kernel machine regression (BKMR) models to explore the mixed relationships between PAHs and parameters of liver function. The present investigation will greatly advance our knowledge of the adverse relationships of PAHs with liver function among pregnant women and provide an important scientific basis for developing effective public health strategies.

## 2. Methods and Materials

### 2.1. Study Population

Pregnant women were drawn from a multi-central birth cohort study conducted between May 2020 and April 2022 in Zunyi City, southwest China. Participants were enrolled from the First Affiliated Hospital of Zunyi Medical University, the Second Affiliated Hospital of Zunyi Medical University, the People’s Hospital of Xishui County, and the People’s Hospital of Meitan County. Women with confirmed pregnancy and ages ≥ 18 years were invited to participate in this study. Each participant was administered a face-to-face questionnaire by uniformly trained investigators. The questionnaire collected general demographic characteristics (such as age, ethnicity, height, weight, etc.), lifestyle (such as passive smoking, exercise frequency, etc.), personal family history (whether suffering from chronic or infectious diseases), and work and family status. This study enrolled women in early pregnancy with the following criteria: (1) Subjects can provide urine and blood samples for analysis; (2) pregnant women with no history of hepatitis; (3) no intention of moving from the residential area within one year. Finally, 294 early pregnant women who had both PAH metabolites and liver function indicators were included in this study. Each participant provided written informed consent after full consideration. The study was approved by the Ethics Committee of the Zunyi Medical University (batch No. [2019] H-005).

### 2.2. Measurement of Urinary PAH Metabolite Concentrations

A polypropylene tube was used to collect 2 mL of urine during the participant’s visit, which was transferred under dry ice and stored frozen in a −80 °C refrigerator as soon as possible until the quantitative analysis. PAH metabolites were determined by gas chromatography-triple quadrupole tandem mass spectrometry (GC-MS, Agilent 7010B; Santa Clara, CA, USA). Briefly, 1.5 mL of urine sample was combined with 1 mL of sodium acetate buffer solution, 10 µL of internal standard solution (50.9 µg/mL of 1-OH-Na-d7 and 5 µg/mL of 1-OH-PH-d9), and 10 µL of glucuronidase/sulfatase and was then hydrolyzed in a water bath at 37 °C for 12 to 16 h. To supersaturate the solution, the hydrolyzed sample was added to MgSO_4_·7H_2_O and combined in a multi-tube vortex mixer for 10 min (2500 r/min). Then, three extractions using n-hexane and ether were carried out. The extracted supernatant was dried with nitrogen after being concentrated with nitrogen to 150 µL. Then, 100 µL of silylation reagent [N. O-bis (trimethylsilane) trifuoroacetamide: trimethylchlorosilane = 99:1] was added to each inner tube and placed in a water bath at 90 °C for 45 min to allow for full derivatization. Finally, the samples were cooled at room temperature and analyzed by gas chromatography-mass spectrometry. Ten PAH metabolites were detected in our study, namely: 1-hydroxyphenanthrene (1-OH-PHE), 2-hydroxyphenanthrene (2-OH-PHE), 3-hydroxyphenanthrene (3-OH-PHE), 4-hydroxyphenanthrene (1-OH-PHE), 9-hydroxyphenanthrene (9-OH-PHE), 2-hydroxyfluorene (2-OH-FLU), 9-hydroxyfluorene (9-OH-FLU), 1-naphthol (1-OH-NAP), 2-naphthol (2-OH-NAP), and 1-hydroxypyrene (1-OH-PYR). Details of PAH metabolite measurement have been described elsewhere [18]. 

The quality control of the PAH measurement is shown in Appendix A, which includes the limit of detection, the limit of quantification, and experimental conditions. The detection limit of 3S/N and the standard addition method were used in this experiment. Three different concentrations of standards were added to a 1.5 mL urine sample and 8 tubes of each concentration were parallel to calculate the recovery rate and precision. A standard curve was made for every 100 samples and converted using the most recent standard curve. In the present study, the standard curve’s coefficient of determination was R^2^ > 0.996. There was at least one parallel sample for every 20 samples to verify the reproducibility of the results. Two parallel samples should have a relative variation of no more than 20%. For each analysis, we monitored the internal intensity, and the internal standard’s response value should be between 70% and 130% of the calibration curve’s response value. Creatinine-corrected concentrations were used to exclude the impact of urine dilution, which was presented in µg/g.

### 2.3. Assessment of Liver Function Markers

ALT and AST are sensitive indicators of liver cell injury or abnormal liver function. ALP is mainly produced in the liver, and pathological elevation mainly occurs in cholestatic liver disease [19]. TBA is the end product of cholesterol catabolism in the liver [20]. The AST/ALT ratio is frequently used to reflect the degree of liver cell damage. 

Venous blood was collected from subjects after at least 8 h fast to test the liver function parameters. A full-automatic hematology analyzer (Sysmex, XN-9000, Sysmex Co., Kobe, Japan) was used to examine liver function parameters, and all the tests were completed by the professional laboratory staff of the Affiliated Hospital of Zunyi Medical University. All testing procedures met quality control requirements.

### 2.4. Covariates

Covariates were determined based on biological possibility and previous research and were obtained from questionnaires completed by face-to-face interviews. Including maternal age (as continuous), pre-pregnancy body mass index (BMI) (as continuous), ethnicity (Han nationality or other), education (<high school, >high school), parity (0, ≥1), passive smoking (yes or no), physical activity (never, 1–2 times/week, ≥3 times/week), household income (<100,000, 100,000–150,000, ≥150,000, and unknown), and gestational hypertension disorder (yes or no). Passive smoking was considered to be passive exposure to cigarettes at least once a month during pregnancy in the workplace, home, or a public place. Weight (in kilos) divided by the square of height (in meters) was used to yield the pre-pregnancy BMI. 

### 2.5. Statistical Analysis

Normal data are shown as means ± standard deviations (SD) and skewed data are expressed as (medians, 25th and 75th percentile), while categorical data are presented as n (%). The distributions of urinary PAHs and liver function parameters are presented as geometric mean (GM) and (medians, 25th and 75th percentile). Concentrations of PAH metabolites, ALT, AST, ALP, TBA, and AST/ALT were subjected to a natural logarithmic transformation due to their skewed distributions. LOD/2 was substituted for all urinary PAH values below the limit of detection (LOD). Additionally, we determined the Pearson correlation coefficients among 10 log-transformed concentrations of PAH metabolites.

Multiple linear regression models were implemented to investigate the individual associations between urinary PAHs and liver function indicators with exposure treated as continuous data and tertiles. 4-OHPHE and 1-OHNAP were modeled as 2-category variables due to their detection rates < 70%. Meanwhile, several sensitivity analyses were performed to demonstrate the robustness of the results. First, we assessed their single associations with all PAHs simultaneously enrolled into multiple linear regression models; second, we excluded subjects with creatinine concentrations outside the range of 0.3–3 g/L. The percent changes were calculated by applying a back transformation of {100 × [exp(β) − 1]} to the corresponding regression coefficients [21]. 

Considering the simultaneous existence and co-exposure of PAHs in the actual environment, we evaluated the mixed associations of PAHs pollutants with liver function parameters using the BKMR model. BKMR can depict interactions between exposure factors as well as complex, nonlinear, and nonadditive exposure–response relationships [22]. We divided PAH contaminants into two categories according to their similar sources of exposure and Pearson correlation values [23,24,25]. We divided 1-OH-PHE, 2-OH-PHE, 3-OH-PHE, 4-OH-PHE, 9-OH-PHE, 1-OH-NAP, 2-OH-NAP, and 1-OH-PYR into group 1 and 2-OH-FLU and 9-OH-FLU into group 2. A hierarchical variable selection technique with 50,000 iterations of the Markov chain Monte Carlo algorithm was used in the model. We determined the group posterior inclusion probability (group PIP) and the conditional posterior inclusion probability (cond PIP) [26,27], and a PIP threshold of 0.5 was considered significant [28].

All models were adjusted for maternal age, pre-pregnancy BMI, ethnicity, education, parity, physical activity, passive smoking, annual household income, and gestational hypertension disorder. All statistical analyses in this investigation were carried out using SPSS version 25.0 and R 4.1.1 software, and all results were statistically significant with the *p* value < 0.05.

## 3. Results

### 3.1. Baseline Characteristics of the Pregnant Woman

A total of 294 early pregnant women were finally included in the present study, and Table 1 presents their basic characteristics. The mean (±SD) of the age and pre-pregnancy BMI was 26.36 ± 4.78 years and 22.50 ± 4.85kg/m^2^, respectively. More than half of pregnant women had an education of less than high school. Pregnant women who are other ethnicities and passive smokers were 0.3% (*n* = 1) and 3.4% (*n* = 10), respectively. Most of the subjects had a reproductive history. Furthermore, 66.7% (*n* = 196) of the women had a family annual income of 100,000–150,000 RMB. 

### 3.2. Distributions of Urinary PAH Metabolites and Liver Function Indicators

The distributions of urinary PAH metabolites and liver function indicators are displayed in Table 2. Urinary PAH metabolites were highly detected with the detection rates of 1-OHPHE, 3-OHPHE, 9-OHPHE, 2-OHFLU, and 9-OHFLU up to 80%. 2-OHNAP had the highest median concentration (0.98 µg/g creatinine), followed by 1-OHNAP (0.57 µg/g creatinine). The Pearson correlation coefficients between 10 log-transformed PAHs metabolites are shown in Appendix A. There was a strong correlation between 2-OHFLU and 9-OHFLU (r = 0.70), while other PAHs were weakly or moderately correlated.

The median concentrations of ALT, AST, ALP, TBA, and AST/ALT were 12.00 (U/L), 18.00 (U/L), 59.50 (U/L), 2.00 (U/L), and 1.42 (U/L), respectively.

### 3.3. Single Associations between Urinary PAH Metabolites and Liver Function Parameters

We evaluated the associations of each PAH metabolite with the liver function parameters using multiple linear regression (Table 3). When PAH metabolites were modeled as continuous variables, each 1% increment in 2-OH-PHE was significantly associated with 3.36% (95% CI: 0.40%, 6.40%) higher ALT and 2.22% (95% CI: 0.80%, 3.67%) higher AST. Each 1% increment in 1-OH-PHE was associated with a 7.04% (95% CI: 1.61%, 12.75%) increased TBA (Table 3). No significant relationships between urinary PAH metabolites and ALP and AST/ALT were found.

Additionally, we analyzed the associations between each PAH metabolite and outcomes with all PAH metabolites simultaneously enrolled in the multiple linear regression models, and the results remained robust (Appendix A).

We excluded subjects with creatinine concentrations outside the range of 0.3 to 3 g/L as a sensitivity analysis to examine the stability of the results [29]. The associations between urinary single PAH metabolites and liver function parameters remained robust after the full adjustment for confounding factors (Appendix A).

When PAHs concentrations were modeled as tertiles, we found that the third tertile of 2-OH-PHE was associated with a 9.31% (95% CI: 0.60%, 18.89%; P-trend = 0.035) higher AST level compared to the first tertile. Similarly, the third tertile of 1-OH-PHE was associated with a 36.75% (95% CI: 9.97%, 70.23%; P-trend = 0.006) increased TBA (Figure 1). 

### 3.4. The Joint Association of PAH Metabolites on Liver Function Based on the BKMR Model

We utilized BKMR, an approach that can address the high dimensional collinearity of multiple chemical exposures in the real world, to explain the overall association of PAHs with liver function parameters (Figure 2). The condpips of 1-OHPHE (0.8218) and 2-OHPHE (0.8947) were the highest in their corresponding groups (Appendix A). PAH mixture was found to be significantly associated with AST when it was kept at the 75th percentile (Figure 2B). When the concentration of PAHs was higher than the 50th percentile, there was an increasing trend in the ALT and TBA models. (Figure 2A,C). The overall associations of PAH mixture with ALP and AST/ALT are shown in Appendix A.

When other PAHs were kept constant at their medians, a positive association of 2-OH-PHE with ALT and AST was observed and 1-OH-PHE was also positively related with TBA (Figure 3). Appendix A showed univariate exposure–response relationships between each PAH metabolite and parameters of ALP and AST/ALT when other chemicals were fixed at their median concentrations.

1-OH-PHE concentration was significantly associated with TBA level when other PAH metabolites were constant at the 50th and 75th percentile (Appendix A).

## 4. Discussion

We assessed the single and joint associations of PAH metabolites with parameters of liver function among early pregnant women. Multiple linear regression models suggested that 1-OH-PHE and 2-OH-PHE were positively correlated with ALT, AST, and TBA, and the BKMR model showed a positive joint association of the PAH mixture with AST. This study expands our understanding of the effect of PAHs on the health of pregnant women.

ALT and AST serve as sensitive indicators of liver cell damage or abnormalities [30], which are often used to diagnose liver dysfunction [31]. It is believed that abnormal liver function during the perinatal period could contribute to adverse effects on maternal and fetal health. Hassen et al. found that higher ALT and AST can be considered biomarkers of organ damage associated with pre-eclampsia in pregnant women [32]. Moreover, premature birth and low birth weight appear more common among mothers with abnormal liver function [8]. Our results indicated that exposure to PAHs may be associated with impaired liver function, as evidenced by increased ALT and AST, especially in women who are more susceptible in early pregnancy. Moreover, in the present study, a positive linear association between 2-OH-PHE and AST was observed. Importantly, the PAH mixture was positively associated with AST, which reflects the joint toxicity of PAH co-exposure. Moreover, a study of 288 petrochemical plant workers found that exposure to PAHs was linked to abnormal ALT (OR = 2.4, 95%CI: 1.2, 4.9) and AST (OR = 4.1, 95%CI: 1.6, 10.2) [33], and similar results have been observed among adolescents [17]. These results imply that the hepatoxicity of PAHs is prevalent across different populations. Notably, in vivo experimental results regarding the generation of PAH hepatoxicity also support our epidemiological findings. ALT and direct bilirubin in adult rats significantly increased after oral administration of pyrene 1500 and 2200 mg/kg for 4 days compared with the control group [9]. In addition, intraperitoneal injection of phenanthrene increased the levels of serum AST and gamma-glutamyl transpeptidase in rats [34], which indicated the toxic effects of PAHs on the liver. 

TBA is the end product of cholesterol catabolism in the liver. Intrahepatic cholestasis of pregnancy can increase the risk of adverse maternal and perinatal outcomes (AMPO). In a study of 1569 Chinese pregnant women, increased TBA concentration was associated with a 30% increased risk of gestational diabetes mellitus and a 22% increased risk of premature rupture of membranes [35]. Our results suggest that exposure to PAHs can increase TBA levels in pregnant women, leading to a possible liver dysfunction. The BKMR model suggested that TBA significantly increased with an increment of 1-OH-PHE concentrations when other PAH metabolites were fixed at the 50th and 75th percentile. To the best of our knowledge, there has been no relevant epidemiological evidence that has uncovered the relationships between PAHs metabolites and TBA. Nevertheless, several animal studies are available to support our results. The TBA in the liver of 4-week-old mice significantly increased after a 3-methylcholanthrene (3MC) treatment for 6 weeks, suggesting that 3MC exposure may interfere with TBA metabolism leading to abnormal enterohepatic circulation and intestinal barrier function [36]. Our study provides new etiological insights regarding PAH-related increased TBA in early pregnancy. Both animal and population studies indicate that exposure to PAHs could result in liver injury, but the hepatoxicity of PAHs on pregnant mice or rats needs further determination. 

The liver, a vital organ for metabolism, is crucial in detoxification. It has been shown that in addition to activating aryl hydrocarbon receptor (AHR) possibly upregulating exogenous metabolic enzymes (e.g., cytochrome P450 monooxygenase 1A1: CYP1A1) to induce their metabolism, some PAHs may also interact with constitutive androstane-receptors (CARs) [37]. The levels of relevant target gene expression (e.g., CYP1A1 and CYP2B6) increased with the aberrant activation of AHR or CAR, which can lead to lipid metabolism disorders, reduced levels of anti-inflammatory elements [38], and a significant decrease in levels of antioxidant enzyme markers (glutathione and catalase) [9]. Reduction in antioxidant enzymes can increase oxidative stress and result in liver damage [39,40]. Therefore, the abnormal processes mentioned above induced by PAHs may contribute to liver dysfunction.

Importantly, extensive studies have been conducted on different species of PAH metabolites and animals to better understand the health effects of PAHs. CYP 1A expression was significantly higher in sea Chelonia exposed to offshore oil spills compared to unexposed birds [41]. In addition, the cytotoxicity, oxidative stress induction, and DNA strand breakage were observed in the liver of rainbow trout after 24 h of PAH exposure (naphthalene, fluoranthene, pyrene, and benzo[a]pyrene: B[a]P) [42]. Collectively, there are multiple molecular signals and multiple pathways that are responsible for PAH-induced liver injury [37]. 

China has a vast territory with a large variation of PAH concentration across the country. The geometric means of PAH metabolites in this study were lower than those in other developed regions [14,43,44,45]. This may be ascribed to the specific environment, air, topography, and traffic in the Guizhou region. Nevertheless, we still observed adverse associations of PAHs with liver function parameters. Therefore, it is necessary to pay attention to the liver function impairment caused by PAHs at higher exposure levels in other regions. Additional investigation is urgently required to determine the connection between PAHs and liver function in pregnant women from different regions. Therefore, our results can supplement the environmental and health significance of PAHs in early pregnancy in southwest China.

Our research has several advantages; firstly, as far as we are aware, we are among the first to explore the potential hepatotoxicity of PAHs in early pregnant women. Secondly, considering the complex PAH exposure in the real world, we utilized the BKMR model to evaluate its health effect as a mixture. However, there are also some limitations in our study. Firstly, because of the cross-sectional nature, it cannot determine the causal link between PAH levels and liver function parameters, and more longitudinal cohort epidemiological studies or animal experiments are needed to confirm our results. Secondly, even if most confounding factors were adjusted, unmeasured or unidentified factors such as PM 2.5, phthalate, diet, and drugs may cause potential confounding effects. Thirdly, a single point urine measurement cannot greatly reflect the long-term PAH levels due to the short PAH half-life. Therefore, future research evaluating PAH exposure with multiple measurements is warranted. Nonetheless, the adverse associations of PAH with the parameters of liver function during early pregnancy observed in this study cannot be ignored.

## 5. Conclusions

We observed that both single PAHs and their mixture were apparently positively correlated with dysregulated liver function, even at low exposure doses. Importantly, this work offers a fresh understanding of the environmental risk factors for abnormal liver function during pregnancy, which could provide target intervention to reduce the risk associated with AMPO. Hence, it is of great significance to monitor the PAH exposure levels and consider an early intervention for the health of pregnant women, which provides an increased chance of a safe maternal pregnancy and safe infant delivery. 

## Figures and Tables

**Figure 1 toxics-11-00863-f001:**
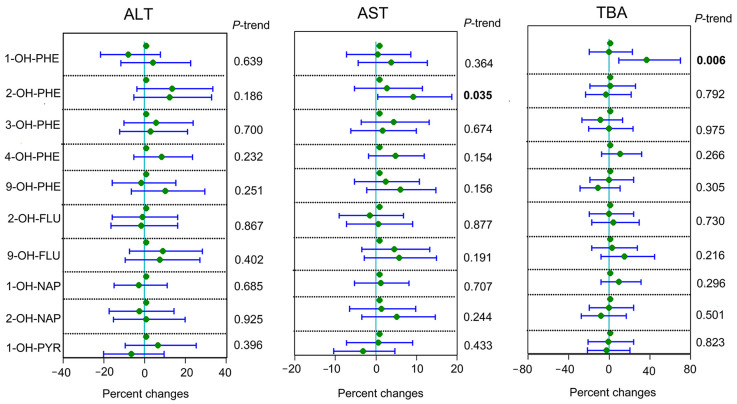
Percent changes for liver function parameters associated with urinary PAH metabolites levels among the study population assessed by linear regression. Adjusted for continuous maternal age, pre-pregnancy BMI, categorical education, parity, ethnicity, passive smoking, household income, and hypertension. Note: the bold to highlight the significant p-trend.

**Figure 2 toxics-11-00863-f002:**
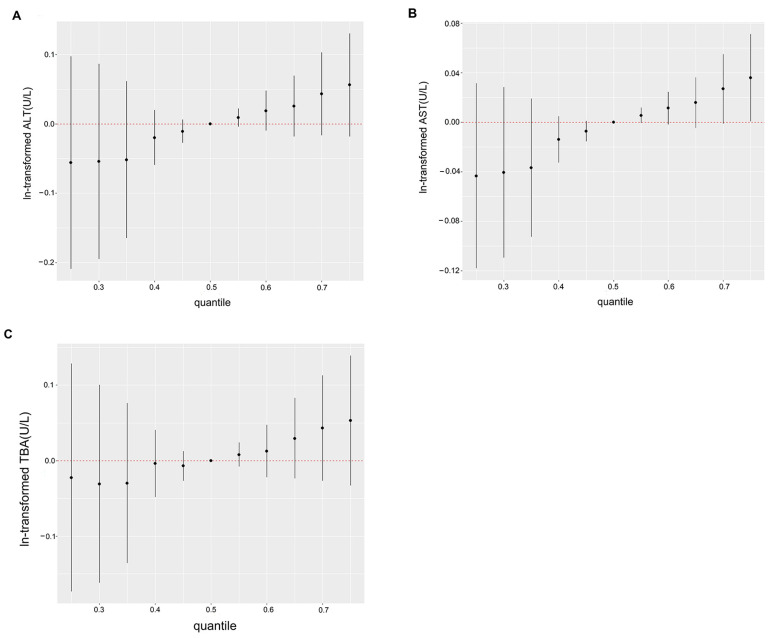
The joint associations of the PAH mixture with ALT (**A**), AST (**B**), and TBA (**C**) levels were estimated by Bayesian Kernel Machine Regression (BKMR). Adjusted for continuous maternal age, pre-pregnancy BMI, categorical education, parity, ethnicity, passive smoking, household income, and hypertension.

**Figure 3 toxics-11-00863-f003:**
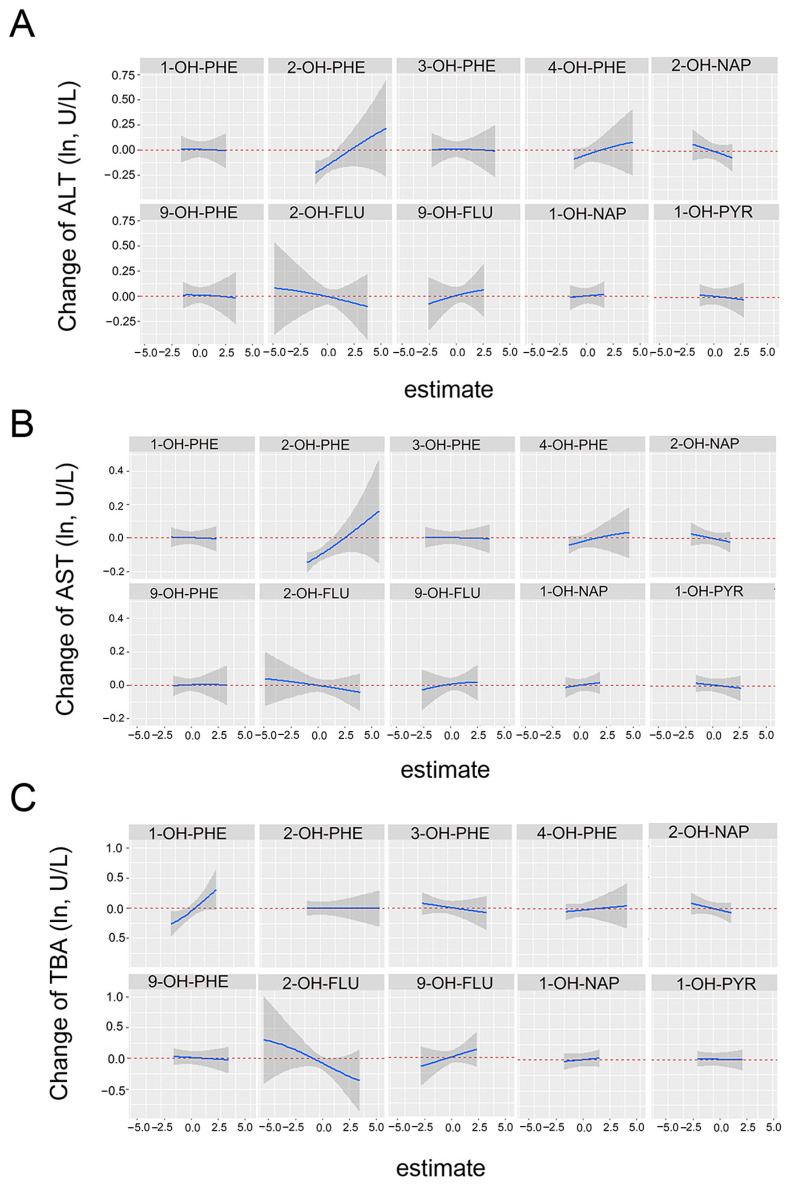
Univariate exposure–response relationship between the concentration of each substance and AST (**A**), ALT (**B**), and TBA (**C**) parameters when other substances were fixed at the median concentration. The model adjusted for continuous maternal age, pre-pregnancy BMI, ethnicity, categorical education, parity, passive smoking, household income, and hypertension.

**Table 1 toxics-11-00863-t001:** Baseline characteristics of the pregnant woman (*n* = 294).

Characteristics	Mean (SD) or *n* (%)
Age (years)	26.36 ± 4.78
Pre-pregnancy BMI	22.50 ± 4.85
Parity	
0	100 (34)
≥1	194 (66)
Education	
Less than high school	171 (58.2)
High school and above	123 (41.8)
Household income (RMB yuan/year)	
<100,000	35 (11.9)
100,000–150,000	196 (66.7)
≥150,000	38 (12.8)
Unknown	25 (8.4)
Passive smoking	
Yes	284 (96.9)
No	10 (3.4)
Physical activity	
Never	122 (41.5)
1–2 times/week	39 (13.3)
≥3 times/week	133 (45.2)
Ethnicity	
The Han nationality	293 (99.7)
Other	1 (0.3)
Gestational hypertension disorder	
Yes	1 (0.3)
No	293 (99.7)
Gestational diabetes mellitus	
Yes	0 (0)
No	294 (100)

Abbreviations: BMI, body mass index; SD, standard deviation.

**Table 2 toxics-11-00863-t002:** Distribution of urinary PAH metabolites and liver function parameters among the study population.

Analyte (µg/g)	Percent Detection	Medians, 25th and 75th Percentile
1-OH-PHE	83.33%	0.09 (0.02, 0.26)
2-OH-PHE	76.53%	0.09 (0.01, 0.55)
3-OH-PHE	92.52%	0.28 (0.09, 0.73)
4-OH-PHE	63.27%	0.03 (0.01, 0.07)
9-OH-PHE	85.37%	0.02 (0.01, 0.05)
1-OH-NAP	64.97%	0.57 (<LOD, 2.05)
2-OH-NAP	79.59%	0.98 (0.05, 4.49)
2-OH-FLU	99.32%	0.55 (0.25, 0.98)
9-OH-FLU	88.78%	0.37 (0.15, 0.80)
1-OH-PYR	71.77%	0.01 (<LOD, 0.07)
Liver function parameters (U/L)		
ALT	-	12.00 (9.00, 18.75)
AST	-	18.00 (15.50, 21.25)
ALP	-	59.50 (47.00, 76.00)
TBA	-	2.00 (1.30, 3.30)
AST/ALT	-	1.42 (1.05, 1.75)

Abbreviations: ALT, alanine aminotransferase; AST, aspartate aminotransferase; ALP, alkaline phosphatase; TBA, total bile acid.

**Table 3 toxics-11-00863-t003:** Relationship between single PAH metabolites in urine and liver function parameters.

Analyte (µg/g)	ALT		AST	
	%Δ (95% CI)	*p*-Value	%Δ (95% CI)	*p*-Value
1-OHPHE	0.70 (−3.15, 4.71)	0.728	0.60(−1.29, 2.63)	0.530
2-OHPHE	3.36 (0.40, 6.40)	0.026	2.22 (0.80, 3.67)	0.003
3-OHPHE	0.90 (−2.86, 4.92)	0.633	0.70 (−1.19, 2.74)	0.456
4-OHPHE	3.67 (−1.98, 9.64)	0.201	2.43 (−0.40, 5.23)	0.097
9-OHPHE	1.82 (−2.86, 6.72)	0.446	1.82 (−0.50, 4.19)	0.131
2-OHFLU	0.20 (−5.26, 5.97)	0.948	0.50 (−2.27, 3.36)	0.711
9-OHFLU	1.41 (−1.88, 4.81)	0.410	1.21 (−0.40, 2.94)	0.153
1-OHNAP	0.50 (−1.69, 2.74)	0.654	0.70 (−0.50, 1.82)	0.244
2-OHNAP	−0.20 (−2.37, 2.12)	0.880	0.40 (−0.80, 1.51)	0.530
1-OHPYR	−0.80 (−3.05, 1.51)	0.492	−0.50 (−1.69, 0.60)	0.356
	ALP		TBA	
	%Δ (95% CI)	*p*-value	%Δ (95% CI)	*p*-value
1-OHPHE	0.10 (−2.76, 3.15)	0.931	7.04 (1.61, 12.75)	0.011
2-OHPHE	−1.49 (−3.63, 0.60)	0.167	0.30 (−3.63, 4.29)	0.900
3-OHPHE	0.70 (−2.18, 3.67)	0.648	−1.98 (−7.04, 3.25)	0.442
4-OHPHE	0.50 (−3.63, 4.81)	0.818	6.08 (−1.69, 14.34)	0.127
9-OHPHE	−1.39 (−4.78, 2.12)	0.441	−1.49 (−7.50, 5.02)	0.650
2-OHFLU	−0.40 (−4.50, 3.87)	0.839	−1.78 (−8.88, 5.87)	0.639
9-OHFLU	−0.70 (−3.15, 1.82)	0.561	2.43 (−2.08, 7.14)	0.287
1-OHNAP	−0.50 (−2.18, 1.11)	0.527	1.21 (−1.78, 4.29)	0.431
2-OHNAP	−1.39 (−3.05, 0.30)	0.114	−1.59 (−4.50, 1.51)	0.314
1-OHPYR	−0.50 (−2.18, 1.21)	0.590	−0.20 (−3.25, 2.94)	0.913
	AST/ALT			
	%Δ (95% CI)	*p*-value		
1-OHPHE	−0.10 (−2.76, 2.74)	0.970		
2-OHPHE	−1.09 (−3.05, 0.9)	0.283		
3-OHPHE	−0.20 (−2.86, 2.53)	0.865		
4-OHPHE	−1.39 (−5.16, 2.63)	0.495		
9-OHPHE	−0.20 (−3.44, 3.15)	0.906		
2-OHFLU	0.30 (3.67, 4.29)	0.866		
9-OHFLU	−0.20 (−2.47, 2.12)	0.885		
1-OHNAP	0.10 (−1.49, 1.61)	0.921		
2-OHNAP	0.60 (−1.00, 2.12)	0.491		
1-OHPYR	0.30 (−1.29, 1.92)	0.730		

Abbreviations: ALT, alanine aminotransferase; AST, aspartate aminotransferase; TBA, total bile acid; ALP, alkaline phosphatase. Percent changes for liver function parameters associated with urinary OH-PAH levels among the study population. Adjusted for continuous maternal age, pre-pregnancy BMI, categorical education, parity, ethnicity, passive smoking, household income, and gestational hypertension disorder.

## Data Availability

The data are available according to request.

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
