# Peer review of "Single and Joint Associations of Polycyclic Aromatic Hydrocarbon Exposure with Liver Function during Early Pregnancy"

_toxics, 2023, doi:10.3390/toxics11100863_

Round 1
Reviewer 1 Report
This study investigated the individual and joint associations of urinary PAHs exposure and liver function during early pregnancy. In general, the study has some novelty, and the results are clearly presented. I have some concerns and suggestions listed in detail as following:
1. The study was conducted during early pregnancy (first-trimester). In the title, it would be more accurate to describe as “during early pregnancy” instead of “during pregnancy”.
2. Diets and drugs taken during early pregnancy should also be considered as confounding factors in the study, as they could potentially influence liver function. It should be listed in limitations of the study.
3. Only AST, ALT, and TBA were analyzed in the BKMR model, what about the other mentioned liver function markers (ALP, AST/ALT)?
4. This is a cross-sectional study that could not make assertions about the causal relationship between PAHs exposure and liver function. Please consider replacing the phrases “combine/joint effects” with “combine/joint association” as the term “effect” typically implies causality.
5. The figures lack numbering, titles and legends. They should be added in the manuscript.:
6. Full name of abbreviations should be provided at their first appearance.
7. Fig.S2, Table S2 and S3, please replace “nation” with “ethnicity” to maintain consistency with the previous description.
The English writing should be improved to some extent.
Reviewer 2 Report
This article measures the relationship between liver function markers and polycyclic aromatic hydrocarbons (PAHs) in pregnant women in Zunyi City, China. Human data is always valuable, and this work is interesting and of potential interest to readers of Toxics. However, some additional information and context is needed. For example, while the authors have indeed identified a correlation between phenanthrene metabolites and AST, it is not clear if any of the measured liver function makers where outside of normal ranges. See my specific comments below. If any of my comments are unclear, please reach out.
Specific Comments
Line 263: Authors assert elevated ALT and AST levels. Typically, clinical evaluation will not diagnose as “elevated” without being outside of normal physiological ranges. As such, I suggest additional context is needed. What are normal ranges for ALT, AST, ALP, TBA, and AST/ALT? Did any of the subjects experience levels outside of normal ranges? Based on distributions in Table 2, I would guess that probably not many if any were outside of normal ranges. This needs to be described in the Discussion and Abstract and authors need to be clear if they state “elevated” it is congruent with clinical diagnosis.
Line 259: how much does ALT and AST need to be elevated for pre-eclampsia?
Line 272 How much elevated?
Line 110: Authors have not provided enough details on their analytical methodology. Authors should provide the extraction method, the column used, GC method used, source of PAH standards and other laboratory supplies, etc.
Line 81: Given should be “given”.
Table 1. Income needs units.
Reviewer 3 Report
Manuscript Toxics 2615815
Title: Single and joint associations of polycyclic aromatic hydrocarbons exposure with liver function during pregnancy
Comments to Authors
General Comments
The manuscript by Dai et al. describes a study aimed at identifying the individual and combined effects of PAH metabolites on liver function during pregnancy. To this end, ten PAH metabolites and five liver function parameters (ALT, AST, ALP, TBA, and AST/ALT ratio) were measured in urine samples collected in a group of 294 first-trimester pregnant women from Southwest China. The association of metabolites with liver function was assessed by multiple linear regression (association of single metabolite with liver function) and by BKMR (joint association of metabolite mixture and liver function).
The current study is on a topic of relevance and general interest for Toxics. The manuscript contributes with new data to better understand the hepatoxicity of PAHs with specific attention to pregnant women for which the topic has not yet explored. The approach adopted for evaluating the association between exposure to PAHs and liver function is adequate for this kind of investigation.
Some issues are however highlighted, mostly on statistical analysis. I’m not an expert in this field but I found difficult to follow the description of the results obtained from statistical evaluation because in my opinion the methods used were not accurately described in the Statistical analysis section.
Overall, the manuscript is generally well written, with well organized introduction and discussion of results. The title of the manuscript reflects the content of the study.
The paper can be accepted after major revision.
Major comments
a) The authors have defined all acronyms in the Abstract but not in the main text. For example, ALP is defined (row 80), but not ALT and AST.
b) Study population: The authors should give a description of the urinary samples collected, if first-morning urine or spot urine samples.
c) Measurements of urinary PAHs Exposure concentrations: The title of this paragraph seems incorrect. The paragraph describes the urinary concentrations of PAH metabolites.
d) Measurements of urinary PAHs Exposure concentrations, line 116: The authors suggest to refer to Tien et al. [18] for details of “PAH measurements”. Has the reference [18] been correctly reported?
e) Measurements of urinary PAHs Exposure concentrations, line 116: The authors should use the term “PAH metabolite measurements” rather than “PAH measurements”, which have not be carried out.
f) Measurements of urinary PAHs Exposure concentrations, line 122: What do the authors mean for “25%, 75%” in parenthesis after the median? The same terms are also repeated later in the text.
g) Measurements of urinary PAHs Exposure concentrations, line 122: The analytical procedure involves the use of internal standard. The authors should indicate which internal standards were used. A very brief description of the analytical procedure in the Supplementary Material may be useful.
h) Measurements of urinary PAHs Exposure concentrations, line 123: It’s a common practice in the biomonitoring studies on PAHs to considered acceptable for statistical evaluation only those samples with urinary creatinine concentrations in the range 0.3–3 g/L (WHO, 1996. Biological Monitoring of Chemical Exposure in the Workplace. Guidelines, vol. 1. Geneva, Switzerlan). Did the authors follow these guidelines?
i) Statistical analysis, line 150: What do the authors mean for “regularly formed variables”?
j) Statistical analysis, line 165: What do the authors mean for “percent changes”? A more detailed description may help.
k) Statistical analysis, lines 178–179: The “group posterior inclusion probability” and the “conditional posterior inclusion probability” terms should be defined to enable the reader to well understand the results described later in the text (a reference may be useful).
l) Results, lines 198-199: Unexpectedly, 2-OH-FLU followed by 9-OH-FLU had the highest concentration. Can the authors suggest an explanation for this result? Accordingly to other studies (Tian et al., 2022; Li et al., Env. Research (2008), 320–321; M. Peng et al., Environmental Pollution 259 (2020) 113854), I would have expected to see urinary concentrations of OH-PAHs inversely related to their molecular weight with higher concentrations for the metabolites of naphthalene.
m) Results, lines 226–228: Please, the authors should clarify the sentence. How they account for the confounding effects of additional PAH metabolites? Were these “additional” metabolites added as independent variables to the model fitted for a given metabolite (dependent variable)?
n) Results, lines 241–245: Please, the authors should better introduce the Figure 2 and 3 (I suppose to be the ones at page 7 and 8), with a short description in the text of what the Figures show.
Minor comments
o) Measurements of urinary PAHs Exposure concentrations, line 123: I suggest using the term “creatinine-corrected concentrations” instead of “Creatinine-calibrated PAHs concentrations”. The authors should also consider that the analytical results are not PAH concentrations but PAH metabolite concentrations.
p) Table 1: The urinary concentrations of OH-PAHs are given in “µg/g” instead of “µg/g creatinine”, as I suppose.
q) Figure at page 7: The number of the reported Figure is missing as well as its captions.
r) Figure at page 8: The number of the reported Figure is missing as well as its captions.
s) Discussion, line 251: “PAH-exposed metabolites”? Perhaps “PAH metabolites”?

Round 2
Reviewer 3 Report
Manuscript Toxics 2615815 R1
Title: Single and joint associations of polycyclic aromatic hydrocarbons exposure with liver function during pregnancy
Comments to Authors
I appreciate authors’ efforts in response to my questions and concerns. The revision clarifies the points I raised but few additional comments/suggestions are provided to improve the quality of the manuscript.
Comments 1: The authors have defined all acronyms in the Abstract but not in the main text. For example, ALP is defined (row 80), but not ALT and AST.
Response 1: Thank you very much for your comments. We have defined all acronyms in the main text when they first appeared and highlighted them in red.
Comments 2: Study population: The authors should give a description of the urinary samples collected, if first-morning urine or spot urine samples.
Response 2: Thanks for the comment. A random spot urine sample was obtained in the field during the participant's visit. A polypropylene tube was used to collect 2 mL of urine, which was transferred under dry ice and stored frozen in a -80°C refrigerator as soon as possible until the quantitative analysis. PAHs metabolites were determined by gas chromatography-triple quadrupole tandem mass spectrometry (GC-MS, Agilent 7010B). Briefly, 1.5 mL of urine sample was combined with 1 mL of sodium acetate buffer solution, 10 uL of internal standard solution, and 10 uL of -glucuronidase/sulphatase and was then hydrolyzed in a water bath at 37 °C for 12 to 16 hours. To supersaturate the solution, the hydrolyzed sample was added to MgSO4-7H2O and combined in a multi-tube vortex mixer for 10 min (2500 r/min). Then, three extractions using n-hexane and ether were carried out. The extracted supernatant was dried with nitrogen after being concentrated with nitrogen to 150 L. 100 µl of silylation reagent was added to each inner tube and placed in a water bath at 90°C for 45 minutes to allow for adequate derivatization. Finally, the samples were cooled at room temperature and analyzed by gas chromatography-mass spectrometry. We have added more details about the urine samples on page 3, lines 109 - 141.
Reviewer: The authors should indicate the silylation reagent used for the derivatization. The derivatization agent used (for example, BSTFA or MSTFA) and experimental conditions adopted may influence on analytical responses.
Comments 3: Measurements of urinary PAHs Exposure concentrations: The title of this paragraph seems incorrect. The paragraph describes the urinary concentrations of PAH metabolites.
Response 3: Thanks a lot and we fully agree with your comments. We have revised it as “Measurement of urinary PAHs metabolites concentrations” and marked it in red (page 3, line 109)
Comments 4: Measurements of urinary PAHs Exposure concentrations, line 116: The authors suggest to refer to Tien et al. [18] for details of “PAH measurements”. Has the reference [18] been correctly reported?
Response 4: Thanks for your comment. We double-checked the standard operating procedure for the detection of urinary PAHs metabolites again, and we verified that the reports in the literature [18] were correct. Additionally, we added more details about the measurements of PAHs metabolites in page 3, lines 113 - 141: The concentrations of 10 PAH metabolites in the urine of pregnant women were determined by gas chromatography-triple quadrupole tandem mass spectrometry (Agilent 7010B). Briefly, 1.5 mL of urine sample was combined with 1 mL of sodium acetate buffer solution, 10 uL of internal standard solution, and 10 uL of -glucuronidase/sulphatase, and hydrolyzed in a water bath at 37 °C for 12 to 16 hours. In order to supersaturate the solution, the hydrolyzed sample was added to MgSO4-7H2O and combined in a multi-tube vortex mixer for 10 min (2500 r/min). Then, three extractions using n-hexane and ether were carried out. The extracted supernatant was dried with nitrogen after being concentrated with nitrogen to 150 L. Add 100 µl of silylation reagent to each inner tube and place in a water bath at 90°C for 45 minutes to allow for adequate derivatization. Finally, the samples were cooled at room temperature and analyzed by gas chromatography-mass spectrometry. We have added more details about the urine samples on page 3, lines 109 - 141.
Reviewer: Thank for your reply. The point was that the paper of Tian et al. [18] was reported by the authors as “Tian Y…. Characteristics of exposure to ….. OCCUP ENVIRON MED. 407 [Journal Article]. 2022 2022-11-24” and I found “Tian Y, et al. Characteristics of exposure to 10 polycyclic aromatic hydrocarbon metabolites among pregnant women: cohort of pregnant women in Zunyi, southwest China. Occup Environ Med 2023; 80:34–41. doi:10.1136/oemed-2022-108324”. They have the same authors, title, and journal but different volume and pages.
Comments 5: Measurements of urinary PAHs Exposure concentrations, line 116: The authors should use the term “PAH metabolite measurements” rather than “PAH measurements”, which have not be carried out.
Response 5: Thanks for your king advice. We have revised it as “PAHs metabolites measurements” and marked them in red (lines 127).
Comments 6: Measurements of urinary PAHs Exposure concentrations, line 122: What do the authors mean for “25%, 75%” in parenthesis after the median? The same terms are also repeated later in the text.
Response 6: Thanks so much for your comment. “25%, 75%” in the paper means 25th and 75th percentile. In statistics, normally distributed data is expressed as mean ± standard deviation, while skewed distributed data is expressed as median and interquartile range and is described as the median (25%, 75%). Because the mean of skewed distributed data can easy be influenced by extreme values, so it is inappropriate to express it as mean ± standard deviation, whereas interquartile range can greatly describe its distribution.
Reviewer: I fully agree with the authors in using percentiles to describe non-normal data distribution but the term “medians (25%, 75%)” is unclear (line 167). In accordance with Table 2, I suggest to modify the text as follows: “medians, 25th and 75% percentile”.
Comments 7: Measurements of urinary PAHs Exposure concentrations, line 122: The analytical procedure involves the use of internal standard. The authors should indicate which internal standards were used. A very brief description of the analytical procedure in the Supplementary Material may be useful.
Response 7: Thank you very much for your helpful advice. The detection limit of 3S/N and the standard addition method were used in this experiment. Three different concentrations of standards were added to a 1.5ml urine sample and 8 tubes of each concentration were parallel to calculate the recovery rate and precision. A standard curve was made for every 100 samples and converted using the most recent standard curve. In the present study, the standard curve's coefficient of determination was R2 > 0.996. There was at least one parallel sample for every 20 samples to verify the reproducibility [1] of the results. Two parallel samples should have a relative variation of no more than [2] 20%. For each analysis, we must monitor the internal intensity, and the internal standard's response value should be between 70% and 130% of the calibration curve's response value. The quality control of PAHs assay is shown in Table S1, which includes the limit of detection, retention time, collision energy, internal standard substance, the limit of quantification, and relative standard deviation. We have added more details on pages 3, lines 130-135.
Reviewer: I appreciate the efforts of the authors to respond my comment but in the text, there is no indication of which substances were used as internal standards (isotopically labeled OH-PAHs? Which ones?)
Comments 8: Measurements of urinary PAHs Exposure concentrations, line 123: It’s a common practice in the biomonitoring studies on PAHs to considered acceptable for statistical evaluation only those samples with urinary creatinine concentrations in the range 0.3–3 g/L (WHO, 1996. Biological Monitoring of Chemical Exposure in the Workplace. Guidelines, vol. 1. Geneva, Switzerlan). Did the authors follow these guidelines?
Response 8: Thank you very much for your comments. We have carefully read this guideline you suggested, and in the guideline it is mentioned that “When the urine is very dilute (relative density < 1.010 or urinary creatinine < 0.3 g/L [ < 2.65 mmol/L]) or concentrated (relative density > 1.030 or urinary creatinine > 3.0 g/L [> 26.5 mmol/L] it is unlikely that any correction will give accurate results”. Nonetheless, a further complication to identifying true exposure levels is that many factors under study might also affect creatinine, such as gender, race, BMI, height, and kidney function. What’s more, creatinine is easily altered by variation in muscle and nutrition, especially the great disparity of nutrition among pregnant women. Therefore, we initially enrolled women whose urinary creatinine outside the range of 0.3 to 3 g/L to further control the possible confounding effect of nutrition of pregnant women. Additionally, to eliminate the possible effect of extreme values of creatinine, we excluded women (n = 7) with urine creatinine concentrations outside the range of 0.3 to 3 g/L as a sensitivity analysis and the results were still robust (Table S3). We have added more details on pages 7, lines 240-243.
Comments 9: Statistical analysis, line 150: What do the authors mean for “regularly formed variables”?
Response 9: Thank you very much for your comments. We are sorry for causing your misunderstanding. We would like to express it as "for normal and skewed distribution data". We have revised it and marked it in red (pages 4, lines 166-167).
Comments 10: Statistical analysis, line 165: What do the authors mean for “percent changes”? A more detailed description may help.
Response 10: Thank you very much for your comments. In the present study the concentrations of PAHs metabolites and parameters of liver function were logarithmically transformed, so it is possible to interpret the specific change in parameters of liver function as a regression coefficient value for each 1-unit increase in lnx. If we want to explain PAHs metabolites, we can say that the change in liver function parameters for every 1%, 10%, etc. increase in PAHs metabolites, at which point the change is log(101/100)*regression coefficient. [A3] In the present study, this is how we describe it: We found that per each 1% increment of urinary 2-hydroxyphenanthrene (2-OH-PHE) was associated with 3.36% (95% CI: 0.40%, 6.40%) higher alanine aminotransferase (ALT) and 2.22% (95% CI: 0.80%, 3.67%) higher aspartate aminotransferase (AST). Per 1% increment in 1-hydroxy-phenanthrene (1-OH-PHE) was significantly associated with 7.04% (95% CI: 1.61%, 12.75%) increased total bile acid (TBA) (pages 1 and 6, lines 21-27 and 225-228). This can be found in the relevant literature and we have inserted it in the main text to facilitate the understanding of relevant terms.[A4] [A5]
Reference: Zota AR, Phillips CA, Mitro SD. Recent Fast Food Consumption and Bisphenol A and Phthalates Exposures among the U.S. Population in NHANES, 2003-2010. Environ Health Perspect. 2016 Oct;124(10):1521-1528. doi: 10.1289/ehp.1510803. Epub 2016 Apr 13. PMID: 27072648; PMCID: PMC5047792.
Comments 11: Statistical analysis, lines 178–179: The “group posterior inclusion probability” and the “conditional posterior inclusion probability” terms should be defined to enable the reader to well understand the results described later in the text (a reference may be useful).
Response 11: Thank you very much for your kind suggestion. Definitions of the terms "group posterior inclusion probability" and "conditional posterior inclusion probability" can be found in the following literature. Moreover, we have inserted relevant literature in the main text to facilitate the understanding of relevant terms.
(1) Bobb JF, Valeri L, Claus Henn B, Christiani DC, Wright RO, Mazumdar M, Godleski JJ, Coull BA. Bayesian kernel machine regression for estimating the health effects of multi-pollutant mixtures. Biostatistics. 2015 Jul;16(3):493-508. doi: 10.1093/biostatistics/kxu058. Epub 2014 Dec 22. PMID: 25532525; PMCID: PMC5963470.
(2) Coull BA, Bobb JF, Wellenius GA, Kioumourtzoglou MA, Mittleman MA, Koutrakis P, Godleski JJ. Part 1. Statistical Learning Methods for the Effects of Multiple Air Pollution Constituents. Res Rep Health Eff Inst. 2015 Jun;(183 Pt 1-2):5-50. PMID: 26333238.
Comments 12: Results, lines 198-199: Unexpectedly, 2-OH-FLU followed by 9-OH-FLU had the highest concentration. Can the authors suggest an explanation for this result? Accordingly to other studies (Tian et al., 2022; Li et al., Env. Research (2008), 320–321; M. Peng et al., Environmental Pollution 259 (2020) 113854), I would have expected to see urinary concentrations of OH-PAHs inversely related to their molecular weight with higher concentrations for the metabolites of naphthalene.
Response 12: Thanks for your comment. We apologize for the data entry error due to our negligence, and we have revised the relevant data in Table 2. We observed the highest geometric concentration of 2-OHFLU (0.51µg/g creatinine), followed by 2-OHNAP (0.35µg/g creatinine). Similarly, in the study of the Tongji birth cohort, we found the highest concentration of 9-OH-FLU, followed by 2-OH-FLU (Luo C, Total Environ. 2022 Dec 15;852:158344). Additionally, according to the study[A6] (Pan Yang, ENVIRON POLLUT. 2018 2018-3-1;234:396-405), we also observed that 1-OHPHE followed by 2-OHNAP had the highest concentration. The relatively low concentration of 1-OHNAP and 2-OHNAP in this study may be due to their relatively low detection rates.
Reference: (1) Yang P, Gong YJ, Cao WC, Wang RX, Wang YX, Liu C, Chen YJ, Huang LL, Ai SH, Lu WQ, Zeng Q. Prenatal urinary polycyclic aromatic hydrocarbon metabolites, global DNA methylation in cord blood, and birth outcomes: A cohort study in China. Environ Pollut. 2018 Mar; 234:396-405. doi: 10.1016/j.envpol.2017.11.082. Epub 2017 Dec 1. PMID: 29202418.
(2) Luo C, Deng J, Chen L, Wang Q, Xu Y, Lyu P, Zhou L, Shi Y, Mao W, Yang X, Xiong G, Liu Z, Hao L. Phthalate acid esters and polycyclic aromatic hydrocarbons concentrations with their determining factors among Chinese pregnant women: A focus on dietary patterns. Sci Total Environ. 2022 Dec 15;852:158344. doi: 10.1016/j.scitotenv.2022.158344. Epub 2022 Sep 2. PMID: 36058337.
Reviewer: I appreciate the efforts to respond my comment but I do not totally agree with the authors. In the Tonji birth cohort (Luo et al., 2022) mentioned by the authors the 2-OH-NAP was reported as the major contributor to OH-PAHs (P50, 1.61 µg/g creatinine), followed by 9-OH-FLU (P50, 0.70 µg/g creatinine), and 2-OH-FLU (P50, 0.53 µg/g creatinine). In addition, Tian et al. (Occup Environ Med 2023;80:34–41. doi:10.1136/oemed-2022-108324) reported that 2-OH-NAP had the highest median concentration (0.50µg/g creatinine), followed by 1-OH-NAP (0.46µg/g creatinine) in the ongoing cohort of pregnant women and neonates in Zunyi (is this the same cohort described in the present manuscript?). Is there something I'm missing?
Comments 13: Results, lines 226–228: Please, the authors should clarify the sentence. How they account for the confounding effects of additional PAH metabolites? Were these “additional” metabolites added as independent variables to the model fitted for a given metabolite (dependent variable)?
Response 13: Thank you for your comment. We analyzed the associations between each PAHs metabolite and outcomes with all PAHs metabolites simultaneously enrolled in the multiple linear regression models, which can control the colinearity among PAHs metabolites(1) and the results remained robust (Table S2). We have added more details on page 7, lines 237-239.
(1) Lai X, Yuan Y, Liu M, Xiao Y, Ma L, Guo W, Fang Q, Yang H, Hou J, Yang L, Yang H, He MA, Guo H, Zhang X. Individual and joint associations of co-exposure to multiple plasma metals with telomere length among middle-aged and older Chinese in the Dongfeng-Tongji cohort. Environ Res. 2022 Nov;214(Pt 3):114031. doi: 10.1016/j.envres.2022.114031. Epub 2022 Aug 4. PMID: 35934145.
Comments 14: Results, lines 241–245: Please, the authors should better introduce the Figure 2 and 3 (I suppose to be the ones at page 7 and 8), with a short description in the text of what the Figures show.
Response 14: Thank you for your suggestion. We have better introduce the Figure 2 and Figure 3 in the text. We have added more details on pages 8 and 9.[A7]
Minor comments
Comments 15: Measurements of urinary PAHs Exposure concentrations, line 123: I suggest using the term “creatinine-corrected concentrations” instead of “Creatinine-calibrated PAHs concentrations”. The authors should also consider that the analytical results are not PAH concentrations but PAH metabolite concentrations.
Response 15: Thanks a lot and we fully agree with your comment. We have replaced the term "creatinine-calibrated PAHs concentrations" with "creatinine-corrected concentrations" (page 3, lines 140). Moreover, we replaced "PAHs" with "PAHs metabolites" in pages6-8, lines 222 and 260.
Comments 16: Table 1: The urinary concentrations of OH-PAHs are given in “µg/g” instead of “µg/g creatinine”, as I suppose.
Response 16: Thanks for your kind advice. We have revised it as “µg/g” (page 6, Table 2).
Comments 17: Figure at page 7: The number of the reported Figure is missing as well as its captions.
Reviewer: Urinary concentrations of OH-PAHs were correctly expressed in µg/g creatinine to take into account urine dilution. The same units should be used in Tables 2 and 3 (not Table 1 as I incorrectly reported).
Response 17: Thanks for your careful review and we are sorry for this mistake. We have made the changes as you suggested which could improve the quality of the paper (page 8).
Comments 18: Figure at page 8: The number of the reported Figure is missing as well as its captions.
Response 18: Thanks for your comment. We have added the number and the captions and we marked them in red (page 8).
Comments 19: Discussion, line 251: “PAH-exposed metabolites”? Perhaps “PAH metabolites”?
Response 19: Thanks for your comment. We have revised it as you suggested, showing on page 8, Line 253.

Round 3
Reviewer 3 Report
I appreciate the efforts that the authors have made in response to my questions and concerns. The revision clarifies the points I raised.